# Evaluating the safety of prenatal HIV PrEP use: Perinatal outcomes from three cohort studies in Western Kenya

Ben Odhiambo[1], Joshua Stern[2], John Kinuthia[1,3], Felix Abuna[1], Eunita Akim[1], Tessa Concepcion[3], Julia C. Dettinger[4], Laurén Gómez[4], Anna Larsen[3], Mary Marwa[1], Jerusha Mogaka[5], Nancy Ngumbau[1], Emmaculate Nzove[1], Barbra A. Richardson[2,4], Salphine Watoyi[1], Grace John-Stewart[2,3,4,6], Jillian Pintye[2,5]*

1 Department of Research and Programs, Kenyatta National Hospital, Nairobi, Kenya, 2 Department of Medicine, University of Washington, Seattle, Washington, United States of America, 3 Department of Global Health, University of Washington, Seattle, Washington, United States of America, 4 Department of Epidemiology, University of Washington, Seattle, Washington, United States of America, 5 School of Nursing, University of Washington, Seattle, Washington, United States of America, 6 Department of Pediatrics, University of Washington, Seattle, Washington, United States of America

* jpintye@uw.edu

## Abstract

Existing data support the safety of daily oral tenofovir disoproxil fumarate (TDF)-based HIV pre-exposure prophylaxis (PrEP) use in pregnancy, yet ongoing monitoring is needed. We analyzed data from three recently completed HIV PrEP safety and implementation studies (PrIMA, PrIMA-X, and mWACh-PrEP) that enrolled women who were offered and/or initiated TDF-based HIV PrEP at routine health clinics in Western Kenya to summarize perinatal outcomes following HIV PrEP use in pregnancy. Data were included in the analysis from participants who were ≥ 15 years, HIV-negative, enrolled ≤ 32 weeks gestation and remained pregnant until at least 24 weeks gestation. We summarized the frequency of each pregnancy outcome (stillbirth, preterm birth, low birthweight, neonatal death, congenital anomalies) by study cohort, HIV PrEP exposure status (any vs. none), and timing of first HIV PrEP exposure (first, second, or third trimester). Poisson regression models were used to assess associations between adverse outcomes and HIV PrEP exposure timing and duration, adjusting for maternal age, primigravity, and clustering by study cohort. A total of N = 4389 women were included in the analysis (29.8% with HIV PrEP exposure). The median age was 24.1 years, and median gestational age at enrollment was 24 weeks. Most women (83.4%) were married and 39.4% had a partner of unknown HIV status. Among HIV PrEP-exposed pregnancies (n = 1310), most initiated HIV PrEP in the second trimester (56.2%). We found no appreciable differences in perinatal outcomes between pregnancies with and without any HIV PrEP exposure, though HIV PrEP-exposed pregnancies had lower frequency of low birthweight (1.9% vs. 2.5%, adjusted prevalence ratio [aPR]=0.77, 95% CI 0.61-0.97). Among pregnancies with any HIV PrEP exposure, preterm birth was less frequent among those with

**Data availability statement:** De-identified and combined data have been made available as Supporting information.

**Funding:** This work was supported by the National Institutes of Health (PrIMA: R01AI125498 to GJS; PrIMA-X: R01HD100201 to JP; and mWACh-PrEP: R01NR019220 to JP and JK). This research was supported by Improving the HIV Care Cascade in Kenya through Implementation Science Training, a grant from the Fogarty International Center, National Institutes of Health D43 TW009580. The funders had no role in study design, data collection and analysis, decision to publish, or preparation of the manuscript.

**Competing interests:** The authors have declared that no competing interests exist.

any PrEP use in the first trimester (aPR = 0.49, 95% CI 0.42-0.57) and third-trimester (aPR = 0.74, 95% CI 0.61-0.88), compared to those with no PrEP use in those trimesters; low birth weight was also less frequent among in pregnancies with third-trimester HIV PrEP initiation compared to second trimester initiation (aPR = 0.74, 95% CI 0.61-0.88). All other perinatal outcomes were comparable by timing of HIV PrEP exposure. These findings support current guidelines recommending daily oral TDF-based HIV PrEP for pregnant and lactating women at risk of HIV.

## Introduction

High HIV incidence among cisgender women of childbearing age remains a public health challenge in HIV high burden countries, including Kenya [1–4]. HIV acquisition risk doubles during pregnancy and postpartum compared to non-pregnant periods [5] due to biological alterations and social and behavioral factors [2,3,5,6]. The World Health Organization (WHO) and Kenya's Ministry of Health recommend offering daily oral tenofovir disoproxil fumarate (TDF)-based pre-exposure prophylaxis (PrEP) for HIV prevention to pregnant and lactating women at substantial ongoing risk for HIV acquisition, based on a large body of safety data on TDF use for HIV treatment among women living with HIV [7–11]. Existing safety data of prenatal HIV PrEP use among women without HIV are reassuring, finding no association between adverse perinatal outcomes and prenatal HIV PrEP use [12]. Yet, safety studies to date focus on women who initiate HIV PrEP during pregnancy, mostly in the second and third tri-mesters only [13–18], and do not evaluate gestational timing or duration of exposure. As oral TDF-based PrEP is scaling up among pregnant and breastfeeding women in East Africa with notable implementation successes in Kenya [12,16], WHO empha-sizes the need for further safety data, especially among women who initiate HIV PrEP outside of research settings. Data from recent HIV PrEP implementation studies among pregnant women could fill remaining safety evidence gaps, especially studies with participants who initiated HIV PrEP in routine settings and prior to pregnancy.

Our team recently conducted three large HIV PrEP safety and/or implementation studies among pregnant women in Western Kenya. The PrIMA Study (PrEP Implementation for Mothers in Antenatal Care; NCT03070600) enrolled $n = 4447$ pregnant participants at any time gestational age and offered HIV PrEP (16.2%% initiated PrEP in pregnancy as part of study procedures) [19]. In addition to prior PrIMA enrollees, the PrIMA Extension Study (PrIMA-X) included a novel observational cohort of $n = 300$ pregnant participants who initiated HIV PrEP before or during pregnancy, yet prior to study enrollment. The mWACh (Mobile Women Adolescent and Child Health; NCT04472884) HIV PrEP Study enrolled $n = 600$ participants who all initiated HIV PrEP between 24–32 weeks gestation within routine antenatal care. We conducted a descriptive prospective analysis using existing data from these studies to summarize the frequency of adverse perinatal outcomes following HIV PrEP exposure and to evaluate timing of HIV PrEP exposure in pregnancy and perinatal outcomes among women in Kenya with and without HIV PrEP exposure.

## Methods

### Study design and participants

This secondary analysis utilized data from participants enrolled in the PrIMA study (http://clinicaltrials.gov/show/ NCT03070600) who did and did not initiate HIV PrEP in pregnancy, novel ANC clients enrolled in the PrIMA-X (PrIMA Extension) study, which was not a clinical trial, who initiated PrEP prior to or during pregnancy, and the mWACh-PrEP study (NCT04472884; all participants-initiated PrEP at 24–32 weeks gestation). We combined cohort data to summarize frequency distributions of adverse perinatal outcomes across cohorts and HIV PrEP exposure statuses. We utilized data collected from all studies up to April 2024 following the inclusion and exclusion criteria for the parent studies with additional criteria (Table 1).

The PrIMA Study was a cluster randomized trial of HIV PrEP counseling strategies conducted between January 2018 and July 2021 in 20 mother and child health clinics in Homa Bay and Siaya counties, Kenya. The study protocol has been described in detail previously [20]. Briefly, antenatal care attendees were eligible for enrollment if they were: currently pregnant, HIV negative, not currently using HIV PrEP, ≥15 years old, tuberculosis negative, planned to reside in the region for at least 1-year postpartum, planned to receive postnatal and infant care at the study facility, and were not currently enrolled in any other studies. Following enrollment, pregnant women were counseled on PrEP as part of routine ANC, either universally (universal arm) or after undergoing HIV risk screening and identified as at risk (targeted arm). Women enrolled at any gestational age during pregnancy and were followed monthly until 9 months postpartum, regardless of HIV PrEP use. For the current analysis, we used data from PrIMA participants who had no history of PrEP use, enrolled

**Table 1. Inclusion and exclusion criteria by study cohort.**

| | PrIMA participants (n = 3595) | PrIMA-X participants (n = 242) | mWACh-PrEP participants (n = 600) |
|---|---|---|---|
| Parent study | **Inclusion** | **Inclusion** | **Inclusion** |
| | • Currently pregnant (any gestational age) | • Currently pregnant (any gestational age) | • Currently pregnant (24–32 weeks gestation) |
| | | | • Initiating HIV PrEP today during ANC |
| | • Not using HIV PrEP at enrollment | • Currently using HIV PrEP | • ≥18 years old |
| | • ≥15 years old | • ≥15 years old | • HIV negative |
| | • HIV negative | • HIV negative | • Receiving MCH services at study sites |
| | • Receiving MCH services at study sites | • Receiving MCH services at study sites | • High HIV risk score (≥6) |
| Current analysis | **Inclusion** | **Inclusion** | **Inclusion** |
| | • Pregnancy outcome data available | • Pregnancy outcome data available | • Pregnancy outcome data available |
| | • HIV PrEP use data available. | • HIV PrEP use data available. | • HIV PrEP use data available. |
| | • Remained pregnant until at least 24 weeks | • Initiated HIV PrEP prior to pregnancy or during pregnancy | **Exclusion** |
| | **Exclusion** | • Remained pregnant until at least 24 weeks | • Multiple pregnancy |
| | • Multiple pregnancy | **Exclusion** | • Seroconverted during pregnancy. |
| | • Seroconverted during pregnancy. | • Multiple pregnancy | • Terminated study during pregnancy. |
| | • Terminated study during pregnancy. | • Seroconverted during pregnancy. | • Missing HIV PrEP use data |
| | • Missing HIV PrEP use data | • Terminated study during pregnancy. | |
| | • Initiated HIV PrEP postpartum | • Missing HIV PrEP use data | |
| | • Pregnancy loss <24 weeks gestation | • Initiated HIV PrEP in 2nd/3rd trimester. | |
| | • Enrolled >32 weeks gestation | • Pregnancy loss <24 weeks gestation | |
| | | • Enrolled >32 weeks gestation | |

HIV PrEP: HIV pre-exposure prophylaxis; MCH: Maternal and child health; ANC: Antenatal care.

in the study at ≤32 weeks gestation and remained pregnant until at least 24 weeks gestation, in order to allow for a non-exposure comparison group of birth outcomes for our other studies.

The PrIMA-X Study is an ongoing longitudinal observational extension cohort study that evaluates safety of oral HIV PrEP use during pregnancy and postpartum at four clinics in Western Kenya. PrIMA-X study follows mother-child pairs up to 60 months post-birth. Women were recruited and enrolled into PrIMA-X study if they were at least 15 years old, HIV negative, and receiving maternal and child health (MCH) services at the participating study sites. The PrIMA-X study population consists of $n=1191$ study participants, of whom $n=891$ are mother-infant pairs rolled over from the PrIMA study. HIV PrEP safety outcome results from the original PrIMA cohort with PrEP exposure have previously been published [19]. The PrIMA-X study newly enrolled $n=300$ pregnant women who initiated HIV PrEP prior to pregnancy or during pregnancy, enrolled at any gestational age, specifically to augment the number of participants with HIV PrEP exposure during pregnancy. For the current analysis, we used data from these $n=300$ newly enrolled PrIMA-X participants who enrolled in the study at ≤32 weeks gestation, initiated HIV PrEP prior to pregnancy or during the first trimester, and remained pregnant until at least 24 weeks gestation.

The mWACh-PrEP study is an ongoing randomized trial conducted at five MCH clinics in Siaya and Kisumu, Kenya, that seeks to improve HIV PrEP adherence using mobile health strategies among women at risk for HIV who initiate HIV PrEP during routine antenatal care. Pregnant women were eligible for enrollment if they were between 24–32 weeks gestation, aged ≥18 years, HIV negative, had an HIV risk score ≥6 (translating to HIV incidence 7.3 per 100 person-years) [21], initiated HIV PrEP that day during ANC, planned to reside in the area for at least one year postpartum, and receive postnatal and infant care at the study clinic. Women who had previously used HIV PrEP were not eligible for enrollment in the study [22]. mWACh-PrEP enrolled $n=600$ women who initiated HIV PrEP during pregnancy.

### Data collection

At enrollment and during study visits, study nurses administered questionnaires using a tablet-based Research Electronic Data Capture (REDCap) [23] in English, Swahili or Dholuo languages as per the woman's preference. Questionnaires include assessment of sociodemographic characteristics, HIV PrEP use information, and obstetric information. Participants self-reported male partner characteristics. Syphilis test results were abstracted from the participant's medical records. The duration of pregnancy was estimated between the first day of the last menstrual period to the date of delivery or pregnancy loss. Evaluation of peripartum outcomes was collected at the first study visit after birth (≤6 weeks) or end of pregnancy for each study participant. All perinatal outcomes (pregnancy loss, stillbirth, preterm birth, low birthweight, congenital anomalies, and neonatal death) were abstracted from clinical records or ascertained by study nurses who are trained in the collection of peripartum outcomes.

### Study measures

Timing of first HIV PrEP exposure in pregnancy was categorized as no HIV PrEP exposure, first trimester (initiated PrEP prior to pregnancy or during the first trimester), second trimester, or third-trimester exposure. Among mWACh-PrEP participants, HIV PrEP initiation date corresponded to the study enrollment date as all participants initiated HIV PrEP same day during ANC as part of the parent study's inclusion criteria. All mWACh-PrEP participants who met the analysis' inclusion criteria were categorized as having second or third-trimester HIV PrEP initiation. Among PrIMA and PrIMA-X participants, HIV PrEP initiation date was ascertained at enrollment and/or follow-up, and timing of HIV PrEP initiation in relation to pregnancy was calculated by subtracting the HIV PrEP initiation date from the estimated pregnancy start date. Outcomes included preterm birth (<37 weeks) determined by last menstrual period, low birth weight (<2500g), stillbirth (pregnancy loss ≥24 weeks), WHO growth indicators (weight-for age Z-score, height-for-age Z-score) [24], congenital malformations, and neonatal death (death of a live-born infant within the first 28 days of life). We did not evaluate small-for-gestational-age due to >50% missingness of birth length. We evaluated each individual outcome in separate models.

## Statistical analysis

Descriptive statistics were used to summarize frequency distributions of adverse pregnancy outcomes. We analyzed and compared each adverse pregnancy outcomes by study cohort, HIV PrEP exposure status, and timing of HIV PrEP exposure (no HIV PrEP exposure, first, second, or third-trimester exposure) using Fisher's exact test for categorical variables (stillbirth, neonatal death, congenital anomalies) and Wilcoxon rank-sum tests to compare the distribution of continuous variables (gestational age, weight at birth) to detect differences between exposure groups since we expect rare outcomes to occur in <5 cases in some groups.

In an exploratory analysis, we used separate Poisson regression models for each individual perinatal outcome to test whether the timing of first HIV PrEP exposure in pregnancy (no HIV PrEP exposure vs. first, second, or third-trimester exposure) was associated with preterm birth, low birth weight, or stillbirth). All models were adjusted for study cohort *a priori*. We also conducted separate exploratory analyses to test whether timing of HIV PrEP initiation (prior to pregnancy vs. during pregnancy) and duration of exposure (time from first PrEP exposure in pregnancy to discontinuation or birth/end of pregnancy) were associated with any adverse outcome. In each model, we accounted for maternal age at enrollment, primigravida, and clustered by study cohort as a random effect. Mode of delivery and infant sex were not included as adjustment variables due to the differential missingness of these variables across cohorts.

## Ethical considerations

Before commencement, the PrIMA, PrIMA-X, and mWACh-PrEP studies received approval from both the University of Washington Institutional Review Board (IRB) and the Kenyatta National Hospital and University of Nairobi Ethics and Research Committee (KNH-UoN ERC): PrIMA (P73/02/2017), PrIMA-X (P921/11/2019), mWACh-PrEP (P319/05/2021). Recruitment occurred from 15 Jan 2018–31 July 2019 for the PrIMA study; 26 Oct 2020–6 June 2023 for the PrIMA-X study; and 14 Feb 2022–13 July 2023 for the mWACh-PrEP study. All women who were interested in participating and met the eligibility criteria, provided written informed consent for their enrollment.

## Results

In total, 4,389 women across the three cohorts met inclusion criteria and were included in the current analysis (82% of the overall combined study population) (Fig 1). The median overall age was 24.1 years (IQR: 21.0,28.6), and the median gestational age at enrollment was 24 weeks (IQR: 20, 28). Most women (83.4%) were married with a median of 10 years of education (IQR: 8.0,12.0). There were differences in the frequency of being primigravida, having a partner known to be living with HIV, and self-reported HIV PrEP adherence across studies (Table 2).

Among women with HIV PrEP exposure during pregnancy ($n=1310$), most initiated HIV PrEP in the second trimester (56.2%), followed by the third trimester (35.5%), first trimester (5.9%), and prior to pregnancy at 2.4%. The median duration of HIV PrEP use during pregnancy was 12.4 weeks (IQR: 8.1,17.0), with participants from the PrIMA-X study reporting the highest median cumulative PrEP use at 20.2 weeks (IQR: 14.4, 27.7). Compared to PrIMA-X (97.9%) and mWACh PrEP (90.4%), the proportion of women in PrIMA still on PrEP at their last pregnancy visit was slightly lower at 84.5%.

Across all cohorts, rates ranged from 1.5-2.7% for stillbirth, 15.1-16.9% for preterm birth, 1.4-3.1% for low birthweight, 9.6-10.1% for low height-for-age Z-score, 2.2-4.6% for low weight-for-age Z-score, 0.4-0.8% for any congenital anomalies, and 1.6-1.7% for neonatal death. Furthermore, there were no significant differences in adverse perinatal outcomes between women without HIV PrEP exposure and any cohort with HIV PrEP exposure (Table 3). Similarly, there were no appreciable differences in the frequency of adverse outcomes by timing of HIV PrEP exposure in pregnancy (Table 4).

In exploratory analyses, there was no statistically significant association between stillbirth occurrence and the timing of first HIV PrEP exposure ($p=0.976$ for the second trimester and $p=0.956$ for the third trimester), cumulative HIV PrEP use ($p=0.883$), or any HIV PrEP exposure during pregnancy ($p=0.617$), compared to no HIV PrEP exposure. Similarly, there was no statistically

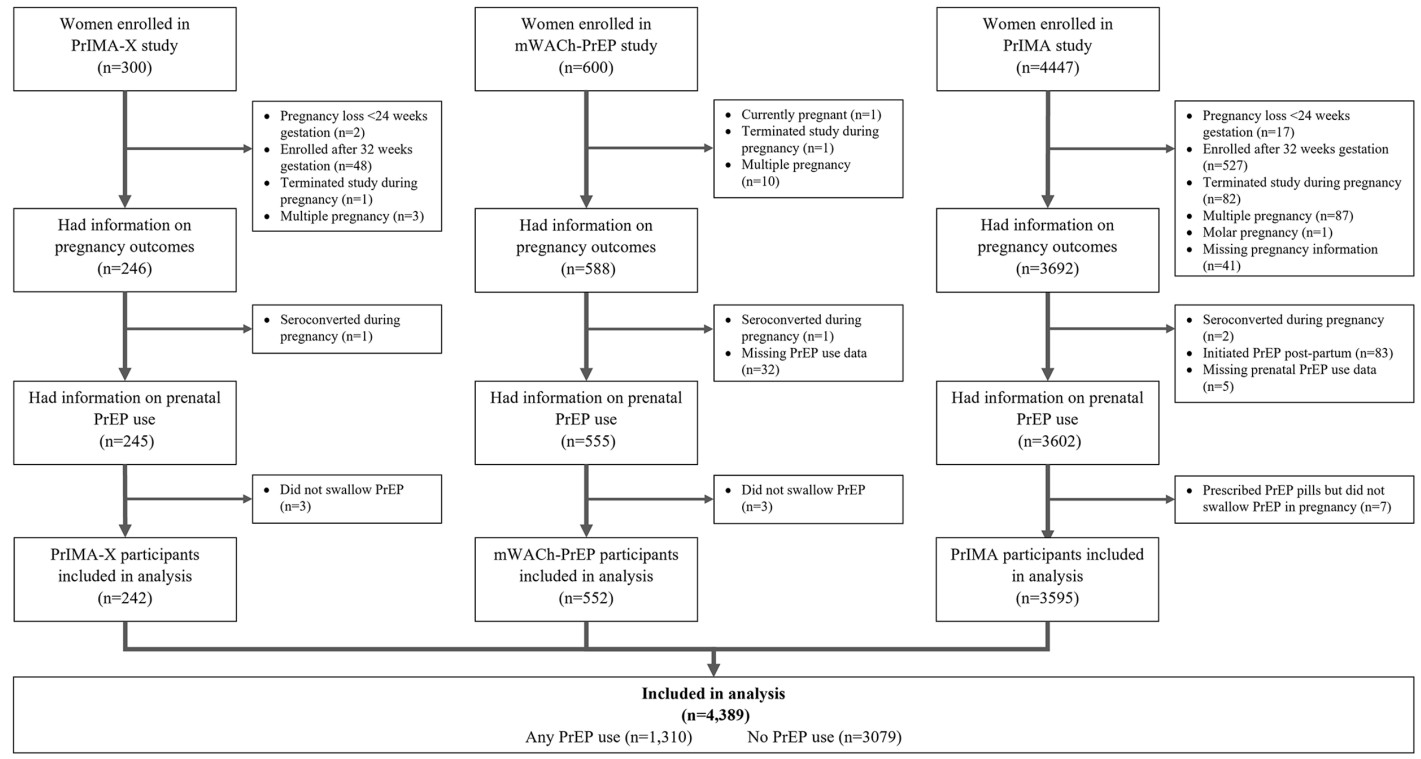

**Fig 1. Flow diagram of enrollment into studies and participant inclusion in combined analyses.**

significant association between the frequency of preterm birth with cumulative HIV PrEP use (p=0.082) or PrEP exposure in the second trimester (p=0.930). However, preterm birth was less frequent among those with HIV PrEP exposure in the first trimester (adjusted prevalence ratio[aPR]=0.49, 95% CI 0.42-0.57, p<0.001) and third trimester (aPR=0.74, 95% CI 0.61-0.88, p=0.001), compared to those without any PrEP exposure in pregnancy. Compared to women without HIV PrEP exposure in pregnancy, the frequency of low birthweight was lower among those with any HIV PrEP exposure in pregnancy (aPR=0.77, 95% CI 0.61-0.97, p=0.027) and those with HIV PrEP exposure in the third trimester (aPR=0.74, 95% CI 0.61-0.88, p=0.001). We did not detect associations between perinatal outcomes and any other HIV PrEP exposure category (**Table 5**).

## Discussion

Our study summarizes safety data for prenatal daily oral HIV PrEP use in ongoing and recently completed studies which include characterization of the timing and duration of HIV PrEP exposure during pregnancy among women in Kenya. Our evaluation contributes novel data from over *n* = 1300 HIV PrEP-exposed pregnancies across three large studies and found no appreciable differences in the frequency of adverse perinatal outcomes between women with and without HIV PrEP exposure. Specifically, rates of stillbirth, congenital malformations, and neonatal death were comparable across HIV PrEP exposure statuses, with less frequent preterm birth and low birth weight among women with HIV PrEP exposure during pregnancy. These findings are consistent with existing safety data, [3,12,15] and extends the evidence base by including data from women who initiated HIV PrEP prior to pregnancy and during the first trimester. Our findings contribute to the growing body of evidence supporting that prenatal HIV PrEP use appears safe among women initiating HIV PrEP outside of research settings, further solidifying that daily oral HIV PrEP is a safe and effective HIV prevention option for pregnant women.

## Table 2. Characteristics of participants included in the analysis (n = 4389).

| Demographics | n (%) or Median (IQR) | | | | |
|---|---|---|---|---|---|
| | Overall (n = 4389) | No HIV PrEP exposure (n = 3079) | Any HIV PrEP exposure | | |
| | | | PrIMA (n = 516) | PrIMA-X (n = 242) | mWACh-PrEP (n = 552) |
| **Women age (years)** | 24.1 (21, 28.6) | 23.9 (20.9, 28) | 25 (21, 30.2) | 25.9 (23.0, 29.9) | 24.8 (21.5, 29.2) |
| <24 years | 2125 (48.4) | 1563 (50.8) | 218 (42.3) | 91 (37.6) | 253 (45.8) |
| 24-35 | 2028 (46.2) | 1383 (44.9) | 257 (49.8) | 125 (51.7) | 263 (47.6) |
| ≥35 years | 230 (5.2) | 131 (4.3) | 41 (8.0) | 22 (9.1) | 36 (6.5) |
| Missing | 6 (0.1) | 2 (0.1) | 0 (0) | 4 (1.7) | 0 (0) |
| **Currently married** | | | | | |
| No | 685 (15.6) | 60 (14.5) | 60 (11.6) | 17 (7.0) | 161 (29.2) |
| Yes | 3662 (83.4) | 2595 (84.3) | 452 (87.6) | 225 (93.0) | 390 (70.7) |
| Missing | 42 (1.0) | 37 (1.2) | 4 (0.8) | 0 (0) | 1 (0.2) |
| **Marriage type** | | | | | |
| Monogamous | 3178 (86.8) | 2298 (88.6) | 339 (75.0) | 192 (85.3) | 349 (89.5) |
| Polygamous | 464 (12.7) | 282 (10.9) | 110 (24.3) | 32 (14.2) | 40 (10.3) |
| Prefer not to answer | 3 (0.1) | 2 (0.1) | 0 (0) | 0 (0) | 1 (0.3) |
| Missing | 17 (0.5) | 13 (0.5) | 3 (0.7) | 1 (0.4) | 0 (0) |
| **Completed education (years)** | 10 (8, 12) | 10 (8, 12) | 9 (8, 12) | 9 (8, 12) | 12 (9, 13) |
| **Partner HIV status** | | | | | |
| HIV-positive | 211 (4.8) | 45 (1.5) | 104 (20.2) | 47 (19.4) | 15 (12.7) |
| HIV negative | 2402 (54.7) | 2080 (67.6) | 200 (38.8) | 110 (45.5) | 12 (2.2) |
| Unknown | 1728 (39.4) | 913 (29.7) | 209 (40.5) | 81 (33.5) | 525 (95.1) |
| No partner/missing | 48 (1.1) | 41 (1.3) | 3 (0.6) | 4 (1.7) | 0 (0) |
| **Pregnancy history** | | | | | |
| **Gestational age at enrollment (weeks)** | 24 (20, 28) | 24 (19, 28) | 24 (18, 28) | 24 (18, 28) | 26 (24, 29) |
| Missing | 0 (0) | 0 (0) | 0 (0) | 0 (0) | 0 (0) |
| **Determination of gestational age** | | | | | |
| LMP | 3843 (87.6) | 3016 (98.0) | 504 (97.7) | 88 (36.4) | 235 (42.3) |
| Fundal height | 503 (11.5) | 63 (2.1) | 12 (2.3) | 146 (60.3) | 282 (51.1) |
| Ultrasound | 0 (0) | 0 (0) | 0 (0) | 8 (3.3) | 35 (6.3) |
| Other | 0 (0) | 0 (0) | 0 (0) | 0 (0) | 0 (0) |
| Missing | 0 (0) | 0 (0) | 0 (0) | 0 (0) | 0 (0) |
| **No. of ANC visits attended to date at enrollment** | 2 (1, 3) | 2 (1, 2) | 1 (1, 2) | 2 (2, 3) | 2 (2, 3) |
| **Primigravida** | | | | | |
| No | 3250 (74.1) | 2244 (72.9) | 438 (84.9) | 217 (89.7) | 351 (63.6) |
| Yes | 1123 (25.6) | 820 (26.6) | 77 (14.9) | 25 (10.3) | 201 (36.4) |
| Missing | 16 (0.4) | 15 (0.5) | 1 (0.2) | 0 (0) | 0 (0) |
| **No. of pregnancies** | 2 (1, 4) | 2 (1,3) | 3 (2, 4) | 3 (2, 4) | 2 (1, 3) |
| **Mode of delivery** | | | | | |
| Vaginal | 3348 (76.3) | 2303 (74.8) | 399 (77.3) | 217 (89.7) | 429 (77.7) |
| C-section | 255 (5.8) | 138 (4.5) | 25 (4.8) | 20 (8.3) | 72 (13.0) |
| Missing/Unknown | 786 (17.9) | 638 (20.7) | 92 (17.8) | 5 (2.1) | 51 (9.2) |
| **Infant sex** | | | | | |
| Male | 1942 (44.3) | 1311 (42.6) | 231 (44.8) | 125 (51.7) | 275 (49.8) |
| Female | 1981 (45.1) | 1374 (44.6) | 225 (43.6) | 113 (46.7) | 269 (48.7) |
| Missing/Unknown | 466 (10.6) | 394 (12.8) | 60 (11.6) | 4 (1.7) | 8 (1.5) |

*(Continued)*

**Table 2.** (Continued)

| Demographics | n (%) or Median (IQR) | | | | |
|---|---|---|---|---|---|
| | Overall (n = 4389) | No HIV PrEP exposure (n = 3079) | Any HIV PrEP exposure | | |
| | | | PrIMA (n = 516) | PrIMA-X (n = 242) | mWACh-PrEP (n = 552) |
| **Birth history among those with prior pregnancies[2]** | | | | | |
| **No. of live births** | 2 (1, 3) | 2 (1, 3) | 2 (1, 3) | 2 (1, 3) | 2 (1, 3) |
| **Total number of living children** | 2 (1, 3) | 2 (1, 3) | 2 (1, 3) | 2 (1, 3) | 2 (1, 2) |
| **Prior miscarriages/ stillbirth** | | | | | |
| No | 2687 (82.7) | 1874 (83.5) | 357 (81.5) | 173 (79.7) | 283 (80.6) |
| Yes | 550 (16.9) | 367 (16.4) | 80 (18.3) | 35 (16.1) | 68 (19.4) |
| Missing | 13 (0.4) | 3 (0.1) | 1 (0.2) | 9 (4.2) | 0 (0.0) |
| **HIV PrEP use** | | | | | |
| **Cumulative PrEP use in pregnancy (weeks)[3]** | 12.4 (8.1, 17) | – | 10.3 (6, 15.6) | 20.2 (14.4, 27.7) | 11.7 (8.3, 15.1) |
| **Timing of PrEP initiation** | | | | | |
| Never initiated PrEP | 3079 (70.2) | 3079 (100) | 0 (0) | 0 (0) | 0 (0) |
| Prior to pregnancy | 32 (2.4) | 0 (0) | 0 (0) | 32 (13.2) | 0 (0) |
| 1st trimester | 77 (5.9) | 0 (0) | 25 (4.8) | 43 (17.8) | 0 (0) |
| 2nd trimester | 736 (56.2) | 0 (0) | 252 (48.8) | 144 (59.5) | 317 (57.4) |
| 3rd trimester | 465 (35.5) | 0 (0) | 239 (46.3) | 23 (9.5) | 235 (42.6) |
| **PrEP status at last ANC visit[4]** | | | | | |
| Never on PrEP | 3079 (70.2) | 3079 (100) | 0 (0) | 0 (0) | 0 (0) |
| On PrEP | 1172 (26.7) | 0 (0) | 436 (84.5) | 237 (97.9) | 499 (90.4) |
| Unknown/missing | 58 (1.3) | 0 (0) | 5 (1.0) | 1 (0.4) | 52 (9.4) |
| Discontinued | 80 (1.8) | 0 (0) | 75 (14.5) | 4 (1.7) | 1 (0.2) |
| **Self-reported PrEP adherence in last 30 days at last ANC visit[5]** | | | | | |
| Never on PrEP | 3079 (71.1) | 3079 (100) | 0 (0) | 0 (0) | 0 (0) |
| No missed pills | 747 (17.3) | 0 (0) | 296 (57.4) | 187 (77.6) | 264 (52.8) |
| Any missed pills | 421 (9.7) | 0 (0) | 140 (27.1) | 48 (19.9) | 233 (46.6) |
| No PrEP use | 81 (1.9) | 0 (0) | 75 (14.5) | 5 (2.1) | 1 (0.2) |
| Missing | 3 (0.1) | 0 (0) | 5 (1.0) | 1 (0.4) | 2 (0.4) |
| **PrEP formulation(s) dispensed during pregnancy** | | | | | |
| Never on PrEP | 3079 (70.2) | 3079 (100) | 0 (0) | 0 (0) | 0 (0) |
| TDF/FTC (300mg/200mg) only | 917 (20.9) | 0 (0) | 182 (35.3) | 233 (96.3) | 502 (90.9) |
| TDF 300mg only | 3 (0.1) | 0 (0) | 3 (0.6) | 0 (0) | 0 (0) |
| TDF/3TC (300mg/300mg) only | 83 (1.9) | 0 (0) | 24 (4.7) | 9 (3.7) | 50 (9.1) |
| Combination | 0 (0) | 0 (0) | 0 (0) | 0 (0) | 0 (0) |
| Missing | 307 (7.0) | 0 (0) | 307 (59.5) | 0 (0) | 0 (0) |

PrEP=pre-exposure prophylaxis for HIV prevention; ANC=antenatal care.

[1]Polygamous marriage was evaluated among those who are married.

[2]Pregnancy and obstetric history assessed among those with history of prior pregnancy at enrollment.

[3]From date of PrEP initiation until date of PrEP discontinuation for mothers who discontinued PrEP use, or pregnancy end date for mothers who continued PrEP.

[4]At last study visit attended during pregnancy after PrEP initiation.

[5]Among women with PrEP status known at last study visit attended during pregnancy. No PrEP use includes women who previously discontinued PrEP.

**Table 3. Birth and neonatal outcomes by HIV PrEP exposure in pregnancy and cohort (n = 3595 from PrIMA; n = 552 from mWACh-PrEP; n = 242 from PrIMA-X).**

| | N | No HIV PrEP exposure[2] (n = 3079) | Any HIV PrEP exposure PrIMA (n = 516) | PrIMA-X (n = 242) | mWACh-PrEP (n = 552) |
|---|---|---|---|---|---|
| | | n (%) or median (IQR) | | | |
| ***Pregnancy and birth outcomes*** | | | | | |
| Any pregnancy loss | 4389 | 64 (2.1) | 14 (2.7) | 4 (1.7) | 8 (1.5) |
| Stillbirth (>24 weeks) | 4389 | 64 (2.1) | 14 (2.7) | 4 (1.7) | 8 (1.5) |
| Gestational age at birth | 4352 | 38 (37, 39) | 38 (37.9, 39) | 38 (38, 39) | 39.4 (37.9, 41) |
| Preterm (<37 weeks) | 4352 | 593 (19.3) | 87 (16.9) | 36 (15.1) | 84 (16.2) |
| Birth weight (kg)* | 2872 | 3.4 (3, 3.6) | 3.3 (3, 3.6) | 3.2 (2.9, 3.5) | 3.2 (2.9, 3.5) |
| Birth weight <2.5 kg* | 2872 | 47 (2.5) | 5 (1.4) | 4 (3.1) | 10 (2.1) |
| Birth length (cm)* | 1403 | 50 (50, 52) | 50 (50, 52) | 50 (49, 50.09) | 49.5 (48, 50) |
| Any congenital anomalies* | 4299 | 23 (0.8) | 4 (0.8) | 1 (0.4) | 2 (0.4) |
| Cleft lip/pallate* | 4299 | 13 (0.4) | 3 (0.6) | 0 (0.0) | 0 (0) |
| Other mouth/gum* | 4299 | 2 (0.1) | 0 (0) | 0 (0) | 0 (0) |
| Club foot* | 4299 | 3 (1.0) | 0 (0) | 0 (0) | 2 (0.4) |
| Jointed fingers or toes* | 4299 | 2 (0.1) | 1 (0.2) | 0 (0) | 0 (0) |
| Extra fingers or toes* | 4299 | 4 (0.1) | 0 (0) | 0 (0) | 0 (0) |
| Missing finger* | 4299 | 0 (0) | 0 (0) | 1 (0.4) | 0 (0) |
| Other limb* | 4299 | 3 (0.1) | 0 (0) | 0 (0) | 0 (0) |
| ***Infant postnatal outcomes*** | | | | | |
| 6-week growth indicators[3]* | | | | | |
| Height-for-age z-score <-2* | 2827 | 182 (9.6) | 27 (10.0) | 22 (10.1) | 40 (9.0) |
| Weight-for age z-score <-2* | 2873 | 56 (2.9) | 6 (2.2) | 10 (4.6) | 17 (3.8) |
| Neonatal death* | 4299 | 43 (1.4) | 8 (1.6) | 4 (1.7) | 9 (1.7) |

*Among 4299 live births.

[1]Includes PrIMA clients who never took PrEP during pregnancy.

[2]Includes new ANC clients from PrIMA-X and mWACh-PrEP client who initiated PrEP the 2nd trimester.

[3]Includes new ANC clients from PrIMA-X and mWACh-PrEP client who initiated PrEP the 3rd trimester.

[4]Includes pregnancy loss, PTB, LBW, SGA, and congenital anomalies.

Most safety studies to date categorize prenatal HIV PrEP exposure as 'any' PrEP exposure during pregnancy versus none, without characterizing variance in timing of exposure. Early discontinuation among women who initiate HIV PrEP during pregnancy and sub-optimal HIV PrEP adherence are well-documented in existing HIV PrEP in pregnancy studies, mainly from Kenya and South Africa [25], signaling that true prenatal exposure is likely highly variable. We did not detect differences in adverse perinatal outcomes by any HIV PrEP exposure status or by timing or duration of HIV PrEP exposure. A recent randomized trial in South Africa comparing immediate HIV oral PrEP initiation among pregnant women at 14–28 weeks gestation to women who deferred initiation of HIV PrEP until breastfeeding cessation found no association between HIV PrEP exposure and preterm birth or small for gestational age [3]. Similarly, two recent studies, one from Kenya and one from South Africa, also found that prenatal HIV PrEP exposure confirmed with quantified tenofovir metabolites in dried blood spots was not associated with adverse pregnancy outcomes, [26,27] though the number of confirmed HIV PrEP-exposed pregnancies was limited in both studies. Our findings add to the limited data on perinatal outcomes following well-characterized timing and duration of HIV PrEP exposure. More studies with sample sizes large enough to

**Table 4. Birth and neonatal outcomes by timing of first HIV PrEP exposure in pregnancy (PrIMA n = 3595; mWACh-PrEP n = 552; PrIMA-X n = 242).**

| | N | n (%) or median IQR | | | |
| --- | --- | --- | --- | --- | --- |
| | | No HIV PrEP exposure[1] (n = 3079) | Any HIV PrEP exposure | | |
| | | | 1st trimester[2] (n = 109) | 2nd trimester[3] (n = 736) | 3rd trimester[4] (n = 465) |
| *Pregnancy and birth outcomes* | | | | | |
| Any pregnancy loss | 4389 | 64 (2.1) | 0 (0) | 16 (2.2) | 10 (2.2) |
| Stillbirth (>24 weeks) | 4389 | 64 (2.1) | 0 (0) | 16 (2.2) | 10 (2.2) |
| Gestational age at birth | 4352 | 38 (37, 39) | 38 (38, 39) | 38.1 (37.6, 40) | 38.3 (38, 40) |
| Preterm (<37 weeks) | 4352 | 593 (19.3) | 10 (9.2) | 134 (18.9) | 63 (13.9) |
| Birth weight (kg)* | 2872 | 3.4 (3, 3.6) | 3.3 (3, 3.5) | 3.3 (3, 3.5) | 3.2 (3, 3.5) |
| Birth weight <2.5 kg* | 2872 | 47 (2.5) | 2 (3.1) | 9 (1.7) | 8 (2.2) |
| Birth length (cm)* | 1403 | 50 (50, 52) | 50 (49, 50.8) | 50 (48.6, 50) | 50 (49, 50.2) |
| Any congenital anomalies* | 4299 | 23 (0.8) | 1 (0.9) | 4 (0.6) | 2 (0.4) |
| Cleft lip/pallate* | 4299 | 13 (0.4) | 0 (0) | 2 (0.3) | 1 (0.2) |
| Other mouth/gum* | 4299 | 2 (0.1) | 0 (0) | 0 (0) | 0 (0) |
| Club foot* | 4299 | 3 (1.0) | 0 (0) | 1 (0.1) | 1 (0.2) |
| Jointed fingers or toes* | 4299 | 2 (0.1) | 1 (0.9) | 0 (0) | 0 (0) |
| Extra fingers or toes* | 4299 | 4 (0.1) | 0 (0) | 0 (0) | 0 (0) |
| Missing finger* | 4299 | 0 (0) | 0 (0) | 1 (0.4) | 0 (0) |
| Other limb* | 4299 | 3 (0.1) | 0 (0) | 0 (0) | 0 (0) |
| *Infant postnatal outcomes* | | | | | |
| 6-week growth indicators[3]* | | | | | |
| Length z-score <-2* | 2827 | 182 (9.6) | 5 (5.2) | 56 (10.5) | 28 (9.3) |
| Weight z-score <-2* | 2873 | 56 (2.9) | 2 (2.1) | 20 (3.7) | 11 (3.6) |
| Neonatal death* | 4299 | 43 (1.4) | 0 (0) | 18 (1.8) | 8 (1.8) |

*Among 4299 live births.

[1]Includes PrIMA clients who never took PrEP during pregnancy.

[2]Includes PrIMA clients and new ANC clients from PrIMA-X who initiated PrEP prior to pregnancy or during the 1st trimester.

[3]Includes PrIMA clients, new ANC clients from PrIMA-X, and mWACh-PrEP clients who initiated PrEP the 2nd trimester.

[4]Includes PrIMA clients, new ANC clients from PrIMA-X, and mWACh-PrEP clients who initiated PrEP the 3rd trimester.

compare pregnancy outcomes following different timing and duration of HIV PrEP use in pregnancy, ideally with quantified HIV PrEP exposure, would help complete the profile of prenatal HIV PrEP use.

We did not detect differences in most perinatal outcomes among women with periconception or first-trimester HIV PrEP exposure compared to those with no exposure or with exposure at later gestational ages. Few data exist on women who became pregnant while on HIV PrEP and are mostly from early HIV PrEP clinical trials in which participants discontinued HIV PrEP upon becoming pregnant [13], therefore limiting HIV PrEP exposure to a very short period near periconception. One small study (n = 35) of women on HIV PrEP who continued use throughout pregnancy found no association between HIV PrEP and adverse perinatal outcomes [15]. To our knowledge, no prior studies to date compare perinatal outcomes among women who initiate HIV PrEP prior to and during pregnancy. The first trimester is a critical window when substantial fetal growth and development occur [25,28], though limited data are available on pregnancy outcomes following first-trimester HIV PrEP exposure. In contrast to safety evaluations of other antiretroviral drugs which suggested that first-trimester exposure might be linked to adverse outcomes compared to exposure later in pregnancy [29,30], our study found no association between HIV PrEP exposure before pregnancy or during the first trimester and adverse perinatal

**Table 5. Association between timing of first HIV PrEP exposure in pregnancy and adverse perinatal outcomes.**

| Adverse perinatal outcomes | Adverse pregnancy outcome | | Univariable Poisson regression | | Multivariable Poisson regression[1] | |
|---|---|---|---|---|---|---|
| | Yes | No | Prevalence Ratio (95% CI) | p-value | Adj. Prevalence Ratio[1] (95% CI) | p-value |
| **Stillbirth (n = 4389)** | 90 (2.1%) | 4299 (98.0%) | | | | |
| PrEP exposure in pregnancy | | | | | | |
| No PrEP exposure | 64 (2.1%) | 3015 (97.9%) | ref | | ref | |
| Any PrEP exposure | 26 (2.0%) | 1284 (98.0%) | 0.95 (0.66-1.38) | 0.804 | 0.91 (0.63-1.31) | 0.617 |
| Cumulative PrEP use in pregnancy – all participants (weeks, n = 4389)[3] | 0 (0, 7) | 0 (0, 6.3) | 0.99 (0.97-1.01) | 0.189 | 0.99 (0.97-1.00) | 0.060 |
| Cumulative PrEP use in pregnancy – among those who used PrEP in pregnancy (weeks, n = 1310)[3] | 11.9 (8.1, 15.1) | 12.4 (8.1, 17) | 0.97 (0.93-1.02) | 0.223 | 0.97 (0.68-1.39) | 0.883 |
| Timing of first PrEP exposure | | | | | | |
| No PrEP exposure | 64 (2.1%) | 3015 (97.9%) | ref | | ref | |
| 1st trimester | 0 (0.0%) | 109 (100.0%) | – | – | – | – |
| 2nd trimester | 16 (2.2%) | 720 (97.8%) | 1.05 (0.79-1.39) | 0.756 | 1.00 (0.74-1.33) | 0.976 |
| 3rd trimester | 10 (2.2%) | 455 (97.9%) | 1.03 (0.56-1.90) | 0.913 | 0.98 (0.52-1.85) | 0.956 |
| **Preterm birth (n = 4352)** | 800 (18.4%) | 3552 (81.6%) | | | | |
| PrEP exposure in pregnancy | | | | | | |
| No PrEP exposure | 593 (19.3%) | 2486 (80.7%) | ref | | ref | |
| Any PrEP exposure | 207 (16.3%) | 1066 (83.7%) | 0.84 (0.81-0.88) | <0.001 | Convergence not achieved | |
| Cumulative PrEP use in pregnancy - all participants (weeks, n = 4352)[3] | 0 (0, 3.5) | 0 (0, 6.8) | 0.98 (0.96-0.99) | 0.003 | 0.98 (0.97-0.99) | 0.003 |
| Cumulative PrEP use in pregnancy – among those who used PrEP in pregnancy (weeks, n = 1273)[3] | 9.6 (6.6, 13.9) | 12.9 (8.4, 17) | 0.96 (0.92-1.00) | 0.064 | 0.96 (0.92-1.01) | 0.082 |
| Timing of first PrEP exposure | | | | | | |
| No PrEP exposure | 593 (19.3%) | 2486 (80.7%) | ref | | ref | |
| 1st trimester | 10 (9.2%) | 99 (90.8%) | 0.48 (0.39-0.58) | <0.001 | 0.49 (0.42-0.57) | <0.001 |
| 2nd trimester | 134 (18.9%) | 575 (81.1%) | 0.98 (0.87-1.11) | 0.766 | 0.99 (0.89-1.11) | 0.930 |
| 3rd trimester | 63 (13.9) | 392 (86.2%) | 0.72 (0.62-0.84) | <0.001 | 0.74 (0.61-0.88) | 0.001 |
| **Low birthweight (n = 2872)[2]** | 66 (2.3%) | 2806 (97.7%) | | | | |
| PrEP exposure in pregnancy | | | | | | |
| No PrEP exposure | 47 (2.5%) | 1847 (97.5%) | ref | | ref | |
| Any PrEP exposure | 19 (1.9%) | 959 (98.1%) | 0.78 (0.60-1.02) | 0.071 | 0.77 (0.61-0.97) | 0.027 |
| Cumulative PrEP use in pregnancy – all participants (weeks, n = 2872)[3] | 0 (0, 4.4) | 0 (0, 8.3) | 0.98 (0.95-1.02) | 0.406 | 0.98 (0.95-1.02) | 0.360 |
| Cumulative PrEP use in pregnancy – only those who used PrEP in pregnancy (weeks, n = 978)[3] | 12.7 (6, 14.4) | 12 (8, 16.3) | 0.99 (0.94-1.05) | 0.817 | 0.99 (0.93-1.05) | 0.800 |
| Timing of first PrEP exposure | | | | | | |
| No PrEP exposure | 47 (2.5%) | 1847 (97.5%) | ref | | ref | |
| 1st trimester | 2 (3.1%) | 62 (96.9%) | 1.26 (0.58-2.75) | 0.563 | 1.26 (0.56-2.84) | 0.584 |
| 2nd trimester | 9 (1.7%) | 533 (98.3%) | 0.67 (0.35-1.28) | 0.222 | 0.65 (0.35-1.21) | 0.179 |
| 3rd trimester | 8 (2.2%) | 364 (97.9%) | 0.87 (0.75-1.00) | 0.044 | 0.85 (0.76-0.94) | 0.001 |

[1]Multivariable Poisson regression adjusted for maternal age at enrollment and primigravida at enrollment; all models accounted for clustering by including study cohort as a random-effect.

[2]Among n = 4299 live births; 66.8% live births were missing birthweight data.

outcomes. Indeed, we found, among women who initiated PrEP early in pregnancy, lower rates of pre-term birth. This observed association may reflect strong health-seeking behaviors and consistent engagement in antenatal care, which could contribute to improved monitoring and better birth outcomes[31,32]. Similarly, women initiating PrEP in late pregnancy (third trimester) were likely to be highly engaged in care during a critical period for fetal growth and preterm birth prevention. Our results provide reassuring evidence regarding the safety of HIV PrEP use during pregnancy.

Our study has some limitations. The mWACh-PrEP study enrolled pregnant women from 24 weeks gestation and subsequently data from the study cannot evaluate the effect of early HIV PrEP exposure on pregnancy losses prior to 24 weeks. To prevent immortal time bias among PrIMA and PrIMA-X participants and to create a comparable cohort, we only included PrIMA and PrIMA-X participants who had a study visit while still pregnant that occurred 24–32 weeks. Therefore, our sample may have a survivorship bias and we could not evaluate early pregnancy loss across cohorts. Additionally, some variables related to perinatal outcomes (e.g., infant sex, birth length, mode of delivery) were incomplete with differential missingness across studies, which limited the outcomes and adjustment variables available for analysis in the dataset. However, data were mostly complete (<10% missingness) for the primary outcomes included. Furthermore, despite the small number of women initiating PrEP in the first trimester or before pregnancy, we found a relatively high prevalence of preterm birth (~20%) in the study population, which yielded 85% power to detect a 10–percentage point difference in preterm birth between women with first-trimester PrEP exposure and those without PrEP exposure. Lastly, the reliance on self-reported data for HIV PrEP adherence may introduce reporting biases. Despite these limitations, our study contributes descriptive data from a large population of HIV PrEP-exposed pregnancies. Although these findings come from research settings, PrEP delivery in these cohorts occurred within routine antenatal care settings where services followed national Ministry of Health guidelines and were integrated into existing clinical workflows. PrEP initiation, adherence, and discontinuation rates observed were comparable to those in the general pregnant population, supporting the real-world relevance and applicability of our findings for broader implementation. At the time of our evaluation, only daily HIV oral PrEP was available outside of research settings in Kenya and therefore we are unable to describe impact of exposure to other HIV PrEP methods. The prospective design and the inclusion of multiple cohorts also strengthen the validity of our results, adding to a growing pool of safety evidence.

## Conclusion

In conclusion, our study found that the frequency of adverse perinatal outcomes was similar among women with and without HIV PrEP exposure during pregnancy, regardless of the timing of HIV PrEP exposure, with lower rates of preterm birth and low birth weight among HIV PrEP-exposed pregnancies. These findings provide reassurance regarding the safety of HIV PrEP use throughout pregnancy and support current guidelines recommending HIV PrEP for pregnant and lactating women at risk of HIV.

## Supporting information

**S1 File. De-identified supporting data file for PrIMA, PrIMA-X and mWACh-PrEP.**
(XLSX)

**S2 File. De-identified supporting data dictionary for PrIMA, PrIMA-X and mWACh-PrEP.**
(XLSX)

## Acknowledgments

We express our sincere gratitude to the dedicated study staff for their tireless efforts and commitment to this research. We are deeply thankful to the participants and their families, healthcare workers, facility leadership and county leadership for their invaluable contributions and trust, without which this study would not have been possible.

## Author contributions

**Conceptualization:** Ben Odhiambo, John Kinuthia, Julia C. Dettinger, Barbra A. Richardson, Grace John-Stewart, Jillian Pintye.

**Data curation:** Ben Odhiambo, Joshua Stern, Laurén Gómez, Mary Marwa, Salphine Watoyi.

**Formal analysis:** Ben Odhiambo, Joshua Stern, Laurén Gómez, Mary Marwa, Salphine Watoyi, Jillian Pintye.

**Funding acquisition:** John Kinuthia, Grace John-Stewart, Jillian Pintye.

**Investigation:** John Kinuthia, Anna Larsen, Barbra A. Richardson, Grace John-Stewart, Jillian Pintye.

**Methodology:** Ben Odhiambo, Joshua Stern, John Kinuthia, Felix Abuna, Eunita Akim, Tessa Concepcion, Julia C. Dettinger, Laurén Gómez, Anna Larsen, Mary Marwa, Jerusha Mogaka, Nancy Ngumbau, Barbra A. Richardson, Salphine Watoyi, Grace John-Stewart, Jillian Pintye.

**Project administration:** Ben Odhiambo, John Kinuthia, Felix Abuna, Eunita Akim, Jerusha Mogaka, Nancy Ngumbau, Emmaculate Nzove, Grace John-Stewart.

**Resources:** Ben Odhiambo, John Kinuthia, Felix Abuna, Eunita Akim, Julia C. Dettinger, Laurén Gómez, Jerusha Mogaka, Nancy Ngumbau, Emmaculate Nzove, Grace John-Stewart, Jillian Pintye.

**Software:** Joshua Stern, Eunita Akim, Laurén Gómez, Mary Marwa, Salphine Watoyi.

**Supervision:** Ben Odhiambo, John Kinuthia, Felix Abuna, Eunita Akim, Julia C. Dettinger, Anna Larsen, Jerusha Mogaka, Nancy Ngumbau, Emmaculate Nzove, Jillian Pintye.

**Validation:** Ben Odhiambo, Tessa Concepcion, Laurén Gómez, Mary Marwa, Salphine Watoyi, Grace John-Stewart, Jillian Pintye.

**Visualization:** Ben Odhiambo, Joshua Stern, Julia C. Dettinger, Laurén Gómez, Anna Larsen, Mary Marwa, Salphine Watoyi.

**Writing – original draft:** Ben Odhiambo, Grace John-Stewart, Jillian Pintye.

**Writing – review & editing:** Ben Odhiambo, Joshua Stern, John Kinuthia, Felix Abuna, Eunita Akim, Tessa Concepcion, Julia C. Dettinger, Laurén Gómez, Anna Larsen, Mary Marwa, Jerusha Mogaka, Nancy Ngumbau, Emmaculate Nzove, Barbra A. Richardson, Salphine Watoyi, Grace John-Stewart, Jillian Pintye.

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
