## [Decision Letter · Decision Letter 0]

17 Oct 2025

PGPH-D-25-02190

Evaluating the Safety of Prenatal HIV PrEP Use: Perinatal Outcomes from Three Cohort Studies in Western Kenya

Dear Dr. Pintye,

Thank you for submitting your manuscript to PLOS Global Public Health. After careful consideration, we feel that it has merit but does not fully meet PLOS Global Public Health’s publication criteria as it currently stands. Therefore, we invite you to submit a revised version of the manuscript that addresses the points raised during the review process.

We look forward to receiving your revised manuscript.

Kind regards,

Everton Falcão de Oliveira, Ph.D

Academic Editor

Journal Requirements:

1. We ask that a manuscript source file is provided at Revision. Please upload your manuscript file as a .doc, .docx, .rtf or .tex.

2. Please provide separate figure files in .tif or .eps format.

3. In the online submission form, you indicated that “Data are available upon reasonable request”.

a. In a public repository,

b. Within the manuscript itself, or

c. Uploaded as supplementary information.

Additional Editor Comments (if provided):

Reviewers' comments:

Reviewer's Responses to Questions

**Comments to the Author**

1. Does this manuscript meet PLOS Global Public Health’s publication criteria?

Reviewer #1: Yes

Reviewer #2: Yes

2. Has the statistical analysis been performed appropriately and rigorously?

Reviewer #1: Yes

Reviewer #2: Yes

3. Have the authors made all data underlying the findings in their manuscript fully available (please refer to the Data Availability Statement at the start of the manuscript PDF file)?

Reviewer #1: Yes

Reviewer #2: Yes

4. Is the manuscript presented in an intelligible fashion and written in standard English?

Reviewer #1: Yes

Reviewer #2: Yes

Reviewer #1: This study is both descriptive and exploratory as seen in the statistical analysis summary.. Poisson regression models were used to assess associations between adverse outcomes and HIV PrEP exposure timing and duration, adjusting for maternal age, primigravity, and clustering by study cohort. The paper is well written and well organized from the statistical perspective. The models for each individual perinatal outcome were presented to test whether the timing of first HIV PrEP exposure in pregnancy (no HIV PrEP exposure vs. first, second, or third-trimester exposure) was associated with preterm birth, low birth weight, or stillbirth). All models were adjusted for study cohort a priori. This allowed , in part, for the combining of the three studies ( which all had sufficient information on pregnancy outcomes and pre natal PrEP use) forming the basis of this retrospective report.

Timing of HIV PrEP initiation and duration of exposure were also associated with any adverse outcome. As is the case in this modeling context, random effect is introduced . The investigators also state that mode of delivery and infant sex were not included as adjustment variables due to the differential missingness of these variables across cohorts.

As noted, the study is primarily descriptive and the major inferential tool as seen in Table 5 is the prevalence ratio (adjusted and unadjusted). The conclusions follow logically from the analyses performed.

The strengths and limitations are well summarized in the discussion. There are a few minor queries.

1. How much missingness of the adjustment variables was there and did it differ by cohort?

2. There is a typo on page 7. The statement, ‘All models adjusted for study cohort a priori.’ should be ‘All models were adjusted for study cohort a priori.’

Reviewer #2: 1. Considering the limited number of first-trimester PrEP initiations relative to later trimesters, can you clarify what analytical approaches were used to address potential power limitations or instability in estimates for this group?

2. Were there any categorical differences between adolescents and adults included in the study?

3. Can you clarify how this study adds to evidence base of initiating PrEP outside of research settings, if all data analyzed was extracted from research studies?

4. Do you have theories on why preterm birth was less frequent among those with exposure in first trimester and third trimester specifically, and similar question on the pattern observed with low birthweight?

**Do you want your identity to be public for this peer review?** For information about this choice, including consent withdrawal, please see our Privacy Policy

Reviewer #1: No

Reviewer #2: No

---

## [Editor Report · Decision Letter 1]

14 Dec 2025

Evaluating the Safety of Prenatal HIV PrEP Use: Perinatal Outcomes from Three Cohort Studies in Western Kenya

PGPH-D-25-02190R1

Dear Dr. Pintye,

We are pleased to inform you that your manuscript 'Evaluating the Safety of Prenatal HIV PrEP Use: Perinatal Outcomes from Three Cohort Studies in Western Kenya' has been provisionally accepted for publication in PLOS Global Public Health.

Best regards,

Everton Falcão de Oliveira, Ph.D

Academic Editor